# Effect of the Soil Matric Potential on the Germination Capacity of *Prosopis chilensis*, *Quillaja saponaria* and *Cryptocarya alba* from Contrasting Geographical Origins

**DOI:** 10.3390/plants11212963

**Published:** 2022-11-02

**Authors:** Ángela Faúndez, Carlos R. Magni, Eduardo Martínez-Herrera, Sergio Espinoza, Suraj Vaswani, Marco A. Yañez, Iván Gréz, Oscar Seguel, Betsabé Abarca-Rojas, Iván Quiroz

**Affiliations:** 1Centro Productor de Semillas y Árboles Forestales, Departamento de Silvicultura y Conservación de la Naturaleza, Facultad de Ciencias Forestales y Conservación de la Naturaleza, Universidad de Chile, Avenida Santa Rosa 11365, La Pintana, Santiago 8831314, Chile; 2Facultad de Ciencias Agrarias y Forestales, Universidad Católica del Maule, Avenida San Miguel 3605, Talca 3466706, Chile; 3Departamento de Ingeniería y Suelos, Facultad de Ciencias Agronómicas, Universidad de Chile, Avenida Santa Rosa 11365, La Pintana, Santiago 8831314, Chile; 4Instituto Forestal and Centro de Investigación de Ecosistemas Mediterráneos (CEIEM), Camino a Coronel Km 7.5, Concepción 4030000, Chile

**Keywords:** soil water content, native flora, germination capacity, seed source

## Abstract

As a consequence of the megadrought in Central Chile, it is expected that most of the distribution of woody species will be narrowed in the northern limits because of restrictions imposed by soil matric potential on seed germination. In this study, we analyzed the effect of the soil matric potential on seed germination and initial recruitment of the sclerophyllous species *Prosopis chilensis*, *Quillaja saponaria* and *Cryptocarya alba* from contrasting geographic origins (i.e., seed sources). We evaluated the germination capacity (%) under different matric potentials (i.e., 0, −6, −33, −750 and −1250 kPa) for 100 days. Soil matric potential of −1250 kPa negatively affected the germination capacity of the three species. *P. chilensis* seeds stopped germinating under soil matric potential close to −1200 kPa, whereas in *Q. saponaria* and *C. alba* the complete inhibition of germination was under −1000 kPa. Seed sources also differed in their germination capacity by soil matric potential: northern seed sources of *P. chilensis* germinated with the lowest soil matric potential. There was no clear trend in *Q. saponaria* and *C*. *alba,* but in general, southern seed sources performed better than the northern ones. The results showed that Ѱ_m_ in the soil played an important role in the germinative capacity against different seed source origins, but not in soils with a north–south gradient.

## 1. Introduction

Plant growth and development depend on climatic factors such as temperature, precipitation, relative humidity, solar radiation, and CO_2_. Thus, climate change may have serious consequences due to the increase in temperatures and especially decreases in rainfall quantity and frequency [1,2], particularly in Mediterranean ecosystems such as the one found in central Chile [3,4,5,6,7,8]. Moreover, under this scenario of climate change, the projections indicate substantial losses of flora diversity in this area [3,9]. In general, the regeneration dynamics of these forests is mainly based on the availability of propagules (i.e., seeds or vegetative structures) [10,11,12,13]. The success of natural regeneration via seeds is generated through the occurrence of a series of processes such as germination, survival, and seedling growth [14]. It is known that both abiotic factors (radiation levels, inadequate temperature, soil water content) and biotic factors (herbivory, competition, insects) affect germination, but in environments with water deficits, abiotic factors are the most influential [15]. However, projected changes in temperature and precipitation regimes, and therefore, in soil moisture will affect many components that determine the success of seeds in the soil.

Tree mortality induced by drought and extreme heat events threatens the provision of forest ecosystem services under the climate change scenario [15,16,17]. The Mediterranean-type climate zone of Chile has experienced a mega-drought since 2010, with a reduction in precipitation of 60% occurring in the warmest decade of the last 100 years [18]. The lower precipitation through consecutive years has decreased soil water availability, affecting the natural regeneration of forest species in this zone [19,20,21], such as the dominant species *Prosopis chilensis* Molina (Stunt), *Quillaja saponaria* Mol., and *Cryptocarya alba* Mol. [7,22]. In this context, the regeneration phase is one of the most critical long-term adaptations of forest species to new environmental conditions [23], while the soil water content is the main abiotic factor determining seed germination and recruitment. Understanding how traits such as germination rate are affected by soil moisture may help the prediction of the impact of climate change on important forest species.

The first germination stage begins with the entry of water into the seed from the external environment (i.e., soil). Then, the hydrated seed (imbibition) activates a series of metabolic processes that are essential for the following stages of the germination process. In soils, water content is typically quantified by the soil matric potential [24] which represents the energy with which water is held by the soil matrix (soil particles and pore space) [25]. It has been observed that a more negative soil matric potential (lower water content) can decrease seed germination [25,26]. Moreover, Gao et al. [26] observed that when the fast drainage pores are drained (0 to 20 kPa), the germination of *Pinus yamannensis* seeds decreases considerably, and that by reducing the water content of the soil to half the usable moisture (750 kPa) the germination capacity in *Bulnesia retama* decreases by 50% compared to seeds that were in a medium with a matric potential at germination capacity at field capacity (33 kPa) [27]. Although the seed germination process is also linked to the plant species’ genetics, within a species the germination capacity may considerably vary among the seed sources (i.e., population geographical origin) [26,28].

In this study, we included three broadly distributed native tree species (*P. chilensis*, *Q. saponaria* and *C. alba*) in Chile, which, according to the predictions, will be differently affected by climate change [3,6]. *P. chilensis* is distributed from north to central Chile (latitudes 24° to 33° S), mainly in the central valley and at coastal sites [29]. According to Miranda et al. [6], *P. chilensis* is considered a phreatophyte species that may have a tap root over 30 m deep with abundant lateral roots, which allows the trees to reach the underground water tables but grows better in sites with shallow water or close to streams. The species’ conservation status is vulnerable because of the degradation of natural populations by overgrazing, wood extraction, and changes in land use. *Q. saponaria* and *C. alba* are two of the most abundant species of the sclerophyllous forest and are distributed from 31° to 37° S, on sites in the Coastal and Andean Mountain range. Compared to the deep-rooted *Q. saponaria*, *C. alba* is considered a hydrophilic shallow-rooted species, so it grows better in areas with high water tables [30]. Regarding the latitudinal distribution and ecological requirements of the species, it is expected that *P. chilensis* and *C. alba* will be the least- and most-affected species by drought events, respectively, in the context of climate change [3]. *P. chilensis* will maintain its current distribution, whereas *Q. saponaria* and *C. alba* will narrow their distributions in the northern part [6]. To contribute information to these predictions, the objectives of this study were: (1) to assess the responses of seed germination to different levels of soil matric potential on *P. chilensis*, *Q. saponaria* and *C. alba*; (2) to assess the variation o those responses among seed sources at the species level; and (3) to determine the relationship between seed germination capacity with climatic and geographic variables associated to the seed source. 

## 2. Results

### 2.1. Germination Capacity at Different Seed Sources and Matric Potentials

There was a significant interaction between the seed source and matric potential treatments for all the species (Table 1, Figure 1, Figure 2 and Figure 3). Contrary to our expectations, none of the species tended to show a latitudinal pattern in germination capacity except *P. chilensis* (Table 2, Figure 2, Figure 3 and Figure 4). In general, *P. chilensis* seed sources had high germination capacity, followed by *C. alba* and *Q. saponaria*. There was no germination in any seed sources for the most restrictive potential of −1250 kPa (Table 1). Only *P. chilensis* presented a continuous decline of the germination capacity with the increase of matric potential (Figure 1). From the northernmost seed source, OV, to the south (LG and CH) there was a steeper decline in the germination capacity as the matric potential increased. In *Q. saponaria* (Figure 2) without restriction at 0 kPa, the highest germination capacity was not obtained, but at −6 and −33 kPa, except in the LB, PO and QL seed sources where it demonstrated a linear behavior (water content decreased and germinative capacity decreased). On average, for all seed sources, the highest germination capacity was obtained with a potential of −6 kPa. In *C. alba* (Figure 3), unlike *P. chilensis* and *Q. saponaria*, there were no significant differences in germination capacity between the matric potentials (0, −6, −33 and −750 kPa) except in *C. alba*, in the seed source HJ, seed sources of the northernmost zone evaluated in this investigation which also had a negative linear behavior.

We also found significant differences in seed weight among the seed sources in all the species (Table 2). The seed weights corresponded to those collected at latitudes of 31° to 34°. This attribute was found to be important in the germination capacity of *C. alba*, but not in the other species. In *C. alba*, seed weight is associated with a higher germination capacity (R^2^ = 0.64, Table 3). The highest variation in seed weight was found in *C. alba* (range from 0.64 g to 1.86 g), followed by *Q. saponaria* (range from 0.06 to 0.18 g).

### 2.2. Critical Soil Matric Potential for Germination

In all the species, there were significant differences among seed sources in the critical matric potential, but this was not attributed to a latitudinal pattern (Table 4). The highest differentiation among seed sources in this parameter was in *Q. saponaria*, followed by *C. alba* and *P. chilensis* (Table 4). However, at the species level, the average critical matric potential of *P. chilensis* was significantly lower than for the other species (−1200 in *P. chilensis* vs. −1036 kPa on average in *Q. saponaria* and *C. alba*).

### 2.3. Environmental Factors Influencing the Germination Capacity

In *P. chilensis*, the health status had a negative influence (higher health status value, worse tree condition) since there is greater germination in trees in worse conditions. With the same influences, such as the Martonne index and precipitation, it had higher germination from sites with more arid climates and less precipitation. In *Q. saponaria*, the health status of the collected trees was what most influenced the germination of this species; trees in better condition have higher germination. In *C. alba*, the most significant influence was the size of the seed: seed weight increased germination in this species.

## 3. Discussion

Regardless of the different soil matric potentials, our results showed a clear effect of seed source on germination capacity, thus confirming the relevance of the geographical origin in the regeneration of the species under study.

In *P. chilensis*, a decrease in the germination capacity was observed with an increase in the soil matric potential of the substrate. The highest germination capacity of the species occurred with a soil matric potential of 0 kPa, which corresponded to the porous system of the saturated substrate. *P. chilensis* seeds require soaking for 48 to 72 h to soften the testa [29], and this was achieved with the saturated substrate. Rodríguez-Rivera et al. [27] evaluated the germinative capacity in *Bulnesia retama* (a species that shares the northern distribution of *Prosopis sp.* in Argentina) under different matric potentials, and together with Varela et al. [31] found that in matric potentials of −1250 kPa, the germination capacity of *B. retama* is practically null. *P. chilensis* exhibited an average germination capacity of 36% at restrictive matric potentials of −1000 kPa. This suggests that the species possess adaptations to low water, which are related to a dormancy mechanism and allow the species to germinate in sites with water limitations in precipitation, and a persistent seed bank that remains dormant until the environmental conditions are conducive to germination [2,3,4,5,6,7,8,9,10,11,12,13,14,15,16,17,18,19,20,21,22,23,24,25,26,27,28,29,30,31,32].

In *Q. saponaria* the highest germination was not obtained with the less restrictive potential, else with intermediate potentials of −6 and −33 kPa. As all seeds and species germination requires gas exchange between the germination medium and the embryo [33], it might be possible that in some seed sources of *Q. saponaria,* the substrate saturated with water (i.e., 0 kPa) does not favor the gaseous exchange for its germination, but it is still close to saturation [31,34]. On seed sources PO, QL and LB, there were a linear response of the germinative capacity with respect to the matric potential of the substrate [24]. With a potential of −750 kPa, which represents half the usable humidity, the germinative capacity of *Q. saponaria* decreased considerably to 28%, as it is a species that does not have special adaptations to water deficits, since it develops in temperate climates [32].

For *C. alba*, the small differences between the matric potentials and the germination capacity might be attributed to the pre-germinative treatment that was applied to remove the pericarp, which presents some chemical inhibitors of germination [32,35]. This caused a homogeneity in the water content of the seed, which generated a more uniform germination capacity in the less restrictive matric potentials (0, −6, −33, and −750 kPa), which may be due to the recalcitrant behaviour of the seeds [36,37,38]. Only the seed source HJ was affected by the matric potential, indicating that the seed source of this species could play an important role in the germination of this species with the use of conventional pregerminative treatments. Saavedra [39] determined the importance in the germination of the seed source of the *C. alba*, the water content of the soil and the positive effect of sowing without pulp. The variability of the seed germinative capacity responds to the specific adaptations to the site where they develop [26,28,39,40,41,42,43].

In *P. chilensis*, we observed a contrasting response of the germination capacity according to the north–south geographic origin. This species may have adaptations to climatic conditions, considering that its largest distribution range is in an arid region [29,32]. The germination capacity was high in seeds from the more arid source OV and LG. According to Martonne’s aridity index (Table 1), seed sources OV and LG are listed as desert. Rodríguez [44] studied the germination capacity of two geographical origins of *Ephedra ochreata* Miers in northern Argentina, a species that shares a distribution with the genus *Prosopis sp.* Among its results, the greatest germination capacity of the species was found in the geographical origins of the most arid areas because of adaptations to xeric sites, resistance to drought, and the permanence of its viability until conditions are favorable [44].

In *Q. saponaria*, there was no clear north–south trend as occurred with *P. chilensis*. The highest germination was found in seed source VP, which comes from a dryland interior area. Its best performance could be associated with the health status of the collected stand compared to the other seed sources, as there was a high correlation between germination capacity and health status. Seed trees with good seed quality are individuals that have a good health status [8,45]. VP was the only stand collected classified as 0 (better health status), compared to ME, LD and BL stands, whose health status was 3 and 2, respectively. The poor health status may have been due to the browning (i.e., wilting) of the stands in these areas because of the uninterrupted mega-drought of the last 12 years [6,46]. This “browning” effect has been observed in seed sources from higher altitudes [2,7]. Thus, the higher germination of seed source VP might be attributed to its lower altitude [6,46,47] better health status [48], and the higher precipitation of the collecting site [7,29].

In *C. alba*, although the southernmost origins where the seeds were collected (i.e., FI, LO and SP) have a lower rainfall deficit, the seed size was the smallest, whereas in those origins with higher germination capacity and major precipitation deficit (i.e., AN and YA), the seeds were larger. Seed size plays an important role in the processes of germination and seedling establishment within a population [49,50], and larger seeds have a higher germination percentage [51], which was observed in our results with *C. alba*. We observed a positive correlation between germination capacity and seed size. In seed sources HJ, SP, LO the lower germination capacity could be attributed to agricultural and forest disturbances rather than low precipitation, which can alter seed quality [52,53] and therefore its health status was not adequate for a seed tree [8]. AN was the seed source with the highest germination capacity, but this stand is under irrigation provided by nearby agricultural and urban plantations, which means that individuals did not suffer the effect of water deficit and the quality of their seeds was not greatly affected [54].

In the critical matric potential, the difference between the species is due to the fact that *P. chilensis* has better adaptability and is better able to germinate in water-restrictive environments (i.e., germination is zero at −1200 kPa) than other species. On the other hand, *Q. saponaria* and *C. alba* are species that share habitats and distributions. Both species develop in Mediterranean-type climates and semi-arid to humid environments and seem to be less adapted to drought (i.e., germination is zero at −1000 kPa) in comparison to *P. chilensis* [55,56,57,58].

Therefore, our results, by including soil variables and their ability to store water so that it is available for germination processes, become relevant. The quality of the soil, especially the water content of the soil, is a key aspect to guarantee the regenerative capacity of the Mediterranean forests of the central zone of Chile [59] and is a very important variable that must be considered in reforestation and restoration plans via direct seeding or when evaluating the regeneration capacity of forests.

## 4. Materials and Methods

### 4.1. Seed Collection and Processing

Seeds were collected between the summer and autumn of 2021 from three geographical origins for *P. chilensis* (from 29° to 32° S), nine for *Q. saponaria* (from 32° to 37° S), and six for *C. alba* (from 32° to 36° S) (Figure 4). The health status of mother trees was determined by a categorical scale (Appendix A) and Martonne climatic index by categorical scale (Appendix A). Seed sources were georeferenced and characterized in soil properties and climatic (Table 5). Seeds were collected directly from 30 trees by seed source. In *P. chilensis*, fruits were opened mechanically to obtain the seeds from the pods. In *Q. saponaria*, the capsules and locules were opened mechanically, separating the seeds by sieving. *C. alba* has recalcitrant type seed, and the collected fruits were immediately stored at 4 °C for a short period of time (1). Moreover, some pre-germination treatments were conducted by species, which correspond to acid scarification for *P. chilensis*, pulp extraction for *C. alba*, and stratification at 4 °C for 7 days for *Q. saponaria*. In *C*. *alba* the pulp was rotted by soaking for 48 h.

#### 4.1.1. Substrate Preparation, Water Retention Curves, and Matric Potential Treatments

The substrate used in this experiment corresponded to an operational mixture of hydrated coconut (80% fiber), perlite (20%), and an initial dose of 1 g L^−1^ of slow-release fertilizer (Basacote 6M, Compo Expert). A preliminary calibration of a water retention curve of the substrate was obtained with subsamples subjected to different matric stresses (i.e., 0, −0.2, −1.0, −6.0 −33, −1000 and −1500 kPa, Appendix A) following the method of [62] using sand bed (and pot and pressure plate (Sandbox for pF determination (pF 0–2.0) Ceramic plates set for pF determination (pF 2.0–4.2)) of Eijkelkamp Soil & Water. Briefly, the freshly prepared substrate was placed on aluminum trays (3240 cm^3^) that were filled to 90% of their capacity with the substrate (approximately 800 g) and was compacted for 30 s by a weight of the same dimensions of 9000 g.

#### 4.1.2. Experimental Design

Based on the calibrated water retention curve, we defined the following matric potential treatments: T1: 0 kPa, T2: −6 kPa, T3: −33 kPa, T4: −750 kPa, T5: −1250 kPa. An additional treatment (T6: −1000) was added for *P. chilensis*. Treatment T3 corresponded to the substrate field capacity. For each species, we followed a design (CRD) with three replications per each combination of seed source and matric potential treatment. Aluminum trays (i.e., repetition) were filled with substrate and prepared according to the described calibration method. Matric potential treatments were reached by drying the substrate at ambient conditions in the laboratory. Then, each aluminum tray was sowed at 0.5 cm depth, and then immediately sealed with Parafilm to avoid water loss. Trays for *Q. saponaria* and *P. chilensis* contained 30 seeds, whereas for *C. alba* 21 seeds. Sowed trays were maintained in a growing chamber at a constant temperature of 16 °C [38]. The trays’ weight was monitored every two days with a precision balance (precision 0.01 g). When necessary, trays were rewetted to maintain the matric potential by adding water until reach the weight associated to each matric potential treatment.

### 4.2. Measurements

Seed variation among seed sources was determined by expressing the average mass of 100 seeds according to ISTA rules [63]. In the germination experiment, a seed was considered germinated with the appearance of the plumule. This was monitored daily for 100 days. At the end of the experiment, the germination capacity was determined as the percentage of germinated seeds over the initial seeds sowed. Dead seeds were also recorded.

### 4.3. Data Analyses

Analysis of variance (ANOVA) on germination capacity (%) was carried out per species, and included the effect of the seed source, matric potential, and the interaction of these two factors in the (CRD). Seed weight was analyzed by a one-way ANOVA, with seed source as main factor. We checked the assumptions of normality of the residuals (Shapiro-Wilk tests) and homoscedasticity (Levene test) accordingly. Post hoc mean comparisons were made with the LSD-Fischer tests. If the errors were not normally distributed, and the variances were not homogeneous, non-parametric tests (Kruskal–Wallis) were performed.

Additionally, the effect of precipitation, aridity index and altitude on the germination capacity was explored through the Pearson’s coefficient of correlation. All the analyses were performed with the Infostat software (2020) and using a significance level of 0.05.

To determine the critical matric potential for germination of each seed source (matric potential where the germination is zero), we fitted a simple linear regression with germination capacity on the y-axis and matric potential in the x-axis and determined the intercept of the model in x-axis.

## 5. Conclusions

Our results showed a clear effect of seed source on germination capacity regardless of the soil matric potentials, thus confirming the relevance of the seed source in the regeneration of the species under study. The matric potential influenced the germination capacity of the evaluated species, being more relevant for *P. chilensis*, where the best germination was found in the saturated medium. In *Q. saponaria*, a saturated medium did not favor germination. *P. chilensis* have a more tolerant strategy as the species can germinate even with restrictive matric potentials, whereas *Q. saponaria* and *C. alba* are more sensitive. In general, no evidence was found on the effect of the environmental conditions analyzed (climatic and geographical) on the germination capacity of the species. In a climate change scenario, the decrease in precipitation may seriously compromise the soil water content and consequently the germination of these species in, especially in the most water-restrictive sites. We advocate that the variation of the seed sources be considered and managed by considering the climate and characteristics of the sites to be restored, and by considering the characteristics of the sites where the seeds are collected.

## Figures and Tables

**Figure 1 plants-11-02963-f001:**
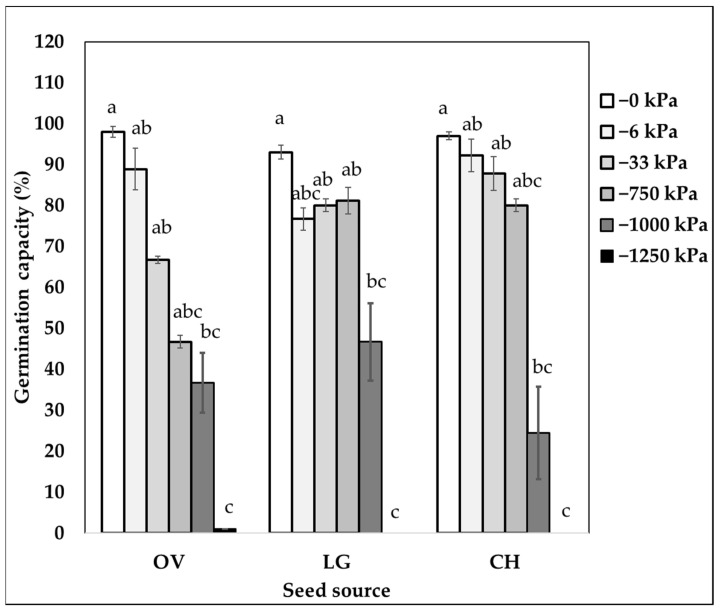
Percentage of germination capacity of *P. chilensis* by seed source and by matric potential (log-kPa). North-south order: OV = Ovalle; LG = Chalinga and CH = Chacabuco. Lowercase letters show a significant difference between matric potential for each seed source according to the non-parametric Kruskal Wallis test (*p* < 0.05).

**Figure 2 plants-11-02963-f002:**
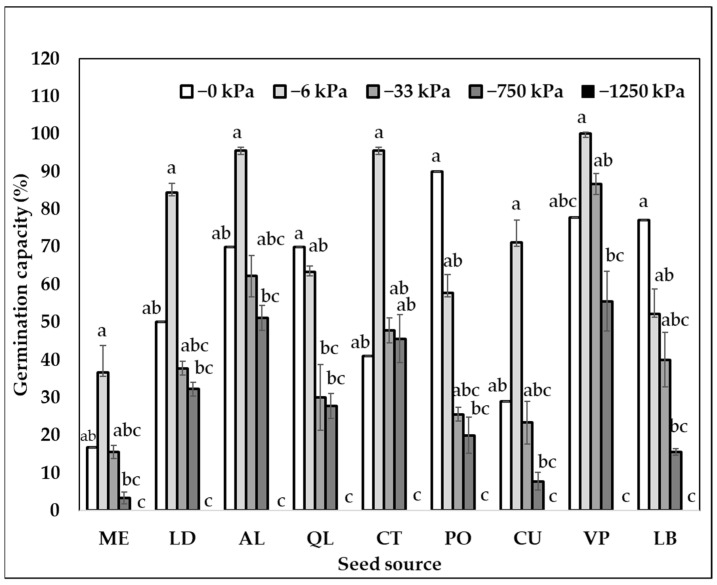
Interaction of germination capacity of *Q. saponaria* by seed source and by matric potential (log-kPa). ME = El Melón; LD = La Dormida; AO = Cantalao; QL = Quebrada de la Plata; CT = Cantillana; PO = Pomaire; CU = Camino el Cobre; VP = Villa Prat and BL = Ñuble. Lowercase letters show a significant difference between matric potential for each seed source according to the non-parametric Kruskal Wallis test (*p* < 0.05).

**Figure 3 plants-11-02963-f003:**
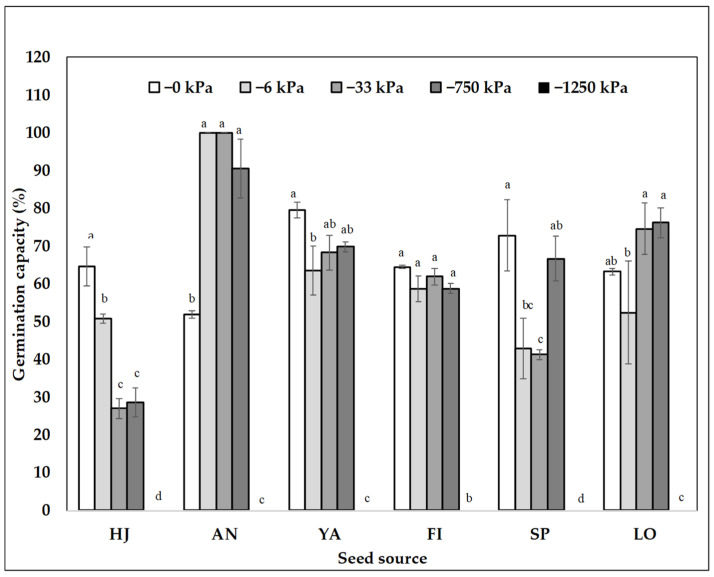
Average of germination capacity of *C. alba* by seed source and by matric potential (log-kPa). North-south order: HJ = Hijuelas; AN = Antumapu; YA = Coya; FI = Infiernillo; SP = San Pedro and LO = Loncomilla. Lowercase letters show significant difference between matric potential for each seed source according to the non-parametric Kruskal–Wallis test (*p* < 0.05).

**Figure 4 plants-11-02963-f004:**
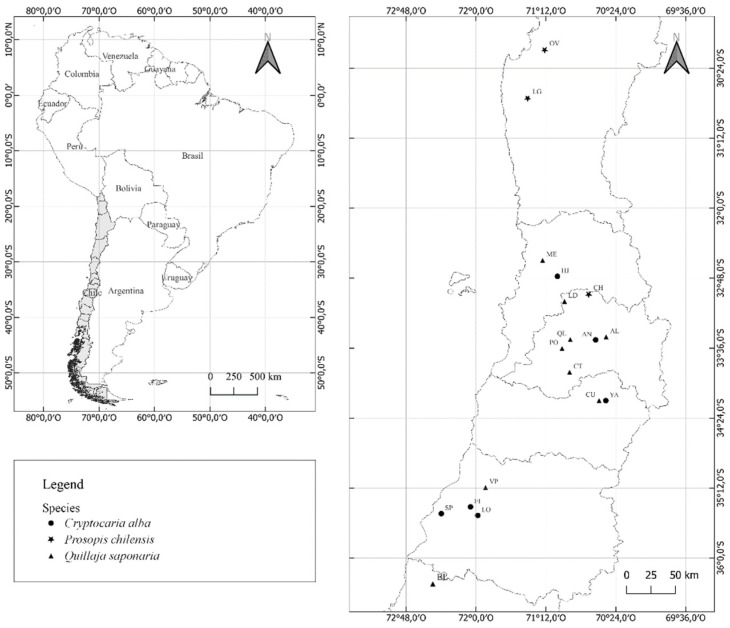
Seed collection sites (seed sources) for *P. chilensis*, *Q. saponaria* and *C. alba* throughout their natural distribution. *P. chilensis*: OV = Ovalle; LG = Chalinga and CH = Chacabuco. *Q. saponaria*: ME = El Melón; LD = La Dormida; AO = Cantalao; QL = Quebrada de la Plata; CT = Cantillana; PO = Pomaire; CU = Camino el Cobre; VP = Villa Prat and BL = Ñuble. *C. alba*: HJ = Hijuelas; AN = Antumapu; YA = Coya; FI = Infiernillo; SP = San Pedro and LO = Loncomilla.

**Table 1 plants-11-02963-t001:** Germination capacity (%) for each species according to matric potential (Ѱ_m_) (*n* = 3). Means, ±SE. Letters indicate significant differences among matric potential in each species for all seed sources. According to ANOVA, and LSD-Fisher multiple comparison test (*p* < 0.05). NA = not measured. (F value: *p* < 0.05 *; *p* < 0.01 *** *p* < 0.001).

	Species		
	*P. chilensis*	*Q. saponaria*	*C. alba*
Ѱ_m_ (kPa)	Germinative Capacity (%)		
**0**	96.0 (±1.2) a	57.9 (±7.8) ab	66.1 (±3.5) a
**−6**	86.3 (±3.9) ab	72.8 (±6.9) a	61.8 (±7.5) a
**−33**	78.2 (±5.0) ab	40.6 (±6.7) b	62.2 (±9.6) a
**−750**	69.3 (±9.2) ab	28.5 (±5.8) c	65.1 (±7.8) a
**−1000**	35.9 (±5.3) bc	NA	NA
**−1250**	0.0 (±0.00) c	0.0 (±0.0) d	0.0 (±0.0) b
***p*-values water potential**	<0.0001	<0.0001	<0.0001
**F-value and significance level**			
**Seed Source (SS)**	3.43 *	55.40 ***	20.75 ***
**Water potential (** **Ѱ_m_)**	133.12 ***	391.40 ***	152.50 ***
**SS × ** **Ѱ_m_**	4.02 ***	12.30 ***	7.22 ***

**Table 2 plants-11-02963-t002:** Mean seed weight and standard error in parenthesis (*n* = 4) per seed source for all the species. Lowercase letters indicate significant differences among seed sources for each species (*P. chilensis*, *Q. saponaria* and *C. alba*) for seed weight, according to non-parametric Kruskal–Wallis test (*p* < 0.05). And capital letters statistically significant differences between species. According to the LSD-Fischer multiple comparison test (*p* < 0.05).

	Geographical Origin	Critical Matric Potential (Ѱm kPa)	Mean for Specie
* **Prosopis chilensis** *	Ovalle (OV)	−1158.1 (±19.8) b	−1191 (±19.9) A
	Chalinga (LG)	−1262.3 (±12.2) a	
	Chacabuco (CH)	−1154.2 (±23.2) b	
* **Quillaja saponaria** *	El Melón (ME)	−901.3 (±109.9) cd	−1010 (±32.8) B
	La Dormida (LD)	−1037.3 (±19.4) bc	
	Cantalao (AL)	−1209.0 (±24.9) ab	
	Quebrada de la Plata (QL)	−1035.7 (±92.0) bc	
	Cantillana (CT)	−964.7 (±32.4) cd	
	Pomaire (PO)	−883.3 (±30.2) cd	
	Camino el Cobre (CU)	−794.3 (±8.7) d	
	Villa Prat (VP)	−1272.0 (±12.7) a	
	Ñuble (LB)	−996.0 (±38.4) c	
* **Cryptocarya alba** *	Hijuelas (HJ)	−1056.8 (±27.3) ab	−1061 (±34.7) B
	Antumapu (AN)	−1021.9 (±25.6) ab	
	Coya (YA)	−1192.6 (±43.9) a	
	Infiernillo (FI)	−1188.3 (±62.6) a	
	Loncomilla (LO)	−1005.6 (±107.4) ab	
	San Pedro (SP)	−904.1 (±59.6) b	

**Table 3 plants-11-02963-t003:** Pearson coefficient of correlation for germination capacity (%) and seed size (g), health status, and climatic (precipitation (mm) and Martonne) and geographical (altitude (m.a.s.l.)) traits. *p*-values in parenthesis.

	Seed Size (g)	Health Status	Precipitation (mm)	Martonne Index	Altitude (m.a.s.l.)
** *P. chilensis* **	−0.36 (0.3306)	0.71 (0.0336)	−0.71 (0.0336)	−0.7 (0.0345)	−0.53 (0.1383)
** *Q. saponaria* **	−0.05 (0.8088)	−0.66 (0.0002)	−0.011 (0.5808)	0.11 (0.5906)	−0.08 (0.6917)
** *C. alba* **	0.8 (0.0001)	−0.21 (0.3954)	−0.42 (0.0845)	−0.41(0.0904)	0.32 (0.2006)

**Table 4 plants-11-02963-t004:** Critical matric potential for per seed source at species level. Lowercase letters indicate statistically significant differences within geographic origins for critical matric potential (kPa) when germinative capacity is 0%. And capital letters statistically significant differences between species. According to the LSD-Fischer multiple comparison test (*p* < 0.05).

	Geographical Origin	Critical Matric Potential (Ѱm kPa)	Mean for Specie
* **Prosopis chilensis** *	Ovalle (OV)	−1158.1 (±19.8) b	−1191 (±19.9) A
	Chalinga (LG)	−1262.3 (±12.2) a	
	Chacabuco (CH)	−1154.2 (±23.2) b	
* **Quillaja saponaria** *	El Melón (ME)	−901.3 (±109.9) cd	−1010 (±32.8) B
	La Dormida (LD)	−1037.3 (±19.4) bc	
	Cantalao (AL)	−1209.0 (±24.9) ab	
	Quebrada de la Plata (QL)	−1035.7 (±92.0) bc	
	Cantillana (CT)	−964.7 (±32.4) cd	
	Pomaire (PO)	−883.3 (±30.2) cd	
	Camino el Cobre (CU)	−794.3 (±8.7) d	
	Villa Prat (VP)	−1272.0 (±12.7) a	
	Ñuble (LB)	−996.0 (±38.4) c	
* **Cryptocarya alba** *	Hijuelas (HJ)	−1056.8 (±27.3) ab	−1061 (±34.7) B
	Antumapu (AN)	−1021.9 (±25.6) ab	
	Coya (YA)	−1192.6 (±43.9) a	
	Infiernillo (FI)	−1188.3 (±62.6) a	
	Loncomilla (LO)	−1005.6 (±107.4) ab	
	San Pedro (SP)	−904.1 (±59.6) b	

**Table 5 plants-11-02963-t005:** Climatic and edaphic characteristics of the seed source. MAT= Mean Annual Temperature (°C); MAP= Mean Annual Precipitation (mm); MAI = Martonne aridity index (1926), S = Total Porosity (%), UW = usable water (%). Health status [48], Deficit pp = precipitation deficit last 12 years (%), ETR = Annual Reference Evapotranspiration (mm); Db = Bulk Density (Mg m^−3^); Dr = Real density (Mg m^−3^), S = Total Porosity (%). Agroclimatic Atlas of Chile, Volume II and III, 2017 [60], Soil samples collected and analyzed at the Soil Physics Laboratory, Faculty of Agronomic Sciences, University of Chile, 2021 [61].

	Geographical Origin	Coordinates UTM		Altitude (masl)	MAT (ºC)	MAP (mm)	MAI	ETR Annual (mm)	Deficit pp (%)	Soil Series	Texture of Soil	Db	Dr	S (%)	Water Retention (kPa)		UW (%)	Health Status
		X	Y									(Mg m^−3^)	(Mg m^−3^)		33	1500		
* **Prosopis chilensis** *	Ovalle (OV)	289,710	6,634,010	446	15.6	66	2.6	1448	89	Tambillo	Sandy loam	1.45	2.71	46	0.13	0.06	7	2
	Chalinga (LG)	269,291	6,596,327	157	18.8	66	2.3	1448	89	Tuqui	Silty Clay Loam	1.62	2.68	40	0.22	0.14	8	2
	Chacabuco (CH)	340,080	6,348,960	641	19.8	259	8.7	1534	75	Rungue	Clayey	1.46	2.65	45	0.31	0.23	8	1
* **Quillaja saponaria** *	El Melón (ME)	289,988	6,391,566	600	14.4	429	17.6	1350	75	Catemu	Loam	1.35	2.65	49	0.2	0.1	10	3
	La Dormida (LD)	314,475	6,340,022	695	14.8	310	12.5	1380	73	Lo Vásquez	Clay Loam	1.45	2.65	45	0.19	0.12	7	1
	Cantalao (AL)	359,604	6,295,762	465	14.7	174	7	1474	72	Asociación Challay	Clay Loam	1.12	2.72	59	0.31	0.18	13	2
	Quebrada de la Plata (QL)	321,508	6,291,796	401	14.8	86	3.5	1489	70	Asociación Munsel	Sandy Clay Loam	1.35	2.52	46	0.29	0.12	17	1
	Cantillana (CT)	321,558	6,250,315	915	14.6	115	4.7	1455	72	Lo Vásquez	Sandy Clay Loam	1.46	2.68	46	0.21	0.09	12	3
	Pomaire (PO)	313,058	6,280,254	216	14.4	407	16.7	1443	58	Pahuilmo	Slimy loam	1.4	2.61	46	0.35	0.25	10	0
	Camino el Cobre (CU)	353,356	621,490	220	14.1	567	23.5	1471	55	Pimpinela	Silty Clay Loam	1.49	2.68	44	0.25	0.13	12	2
	Villa Prat (VP)	236,739	6,101,855	404	14.5	671	27.4	1470	38	Lontue	Sandy loam	1.25	2.65	53	0.23	0.14	9	3
	Ñuble (LB)	725,000	5,980,500	945	13.9	858	35.9	1371	33	Asociación Treguaco	Sandy Clay Loam	1.26	2.65	52	0.26	0.14	12	2
* **Cryptocarya alba** *	Hijuelas (HJ)	308,503	6,260,416.3	473	14.4	429	17.6	1350	71	Ocoa	Sandy loam	1.18	2.49	53	0.25	0.15	10	0
	Antumapu (AN)	348,515	6,291,391	629	14.7	174	7	1474	68	Santiago	Clay loam	1.15	2.57	55	0.34	0.19	15	0
	Coya (CY)	360,498	6,214,498	1047	14.1	567	23.5	1471	55	Asociación Sierra Bellavista	Sandy loam	1.45	2.67	46	0.22	0.13	9	1
	Infiernillo (FN)	221,871	6,076,473	150	14.5	671	27.4	1470	40	Asociación Pocillas	Loamy Clay Loam	1.41	2.62	46	0.27	0.16	11	2
	Loncomilla (LN)	229,719,9	6,065,850	88	13.8	720	30.3	1521	33	Asociación Cauquenes	Clay loam	1.45	2.69	46	0.25	0.16	9	1
	San Pedro (SP)	191,758	6,066,938	230	13.3	920	39.5	1271	35	Asociación Constitución	Sandy Clay Loam	1.37	2.58	47	0.31	0.21	10	3

## Data Availability

Not applicable.

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
