# Peer review of "Effect of the Soil Matric Potential on the Germination Capacity of Prosopis chilensis, Quillaja saponaria and Cryptocarya alba from Contrasting Geographical Origins"

_plants, 2022, doi:10.3390/plants11212963_

Round 1
Reviewer 1 Report
This manuscript deals with the effect of soil water availability and seed germination of several seed sources from three species. The topic is within the scope of the journal, is interesting and original. The manuscript is very well structured and the bibliography is very up-to-date. However, there are some aspects of the manuscript, such as format errors and mainly Results presentation that should be clarified. All this has been highlighted in yellow in the attached file.
I cannot recommend this manuscript for publication in the present form but could be accepted after a major revision.

Author Response
Dear, thank you very much for your corrections, I attach the document with the suggested corrections.
Hugs

Reviewer 2 Report
1. introduction : The plant growth and development depend on climatic factors such as temperature, precipitation, relative humidity, solar radiation, and CO2. Thus, climate change may have serious consequences [1, 2]
--> It is true that plant growth is affected by climate change. However, if it is stated in the very next sentence that it has serious consequences, it seems to be too generalized. it need an explanation as to why this is such a serious outcome.
2. line 37 : under this scenarios --> Exactly what scenario does this mean?
3. line 77-78 : According to [48], P. chilensis is considered a phre ~~
--> need to change to like "according to authors [48]" ~~
4. line 108-109 : ", except in the LB origin where it had a linear behavior (water content decreases and germinative capacity decreases)."
--> PO also showed a similar pattern to LB.
5. table 1 : There are two duplicate parentheses. delete it
6. Change commas to decimal points in tables. Other tables also need corrections.
7. In Figure 2 and Figure 3, add the botanical name. In other words, it needs to be supplemented so that readers can understand it just by looking at the title of the figures.
8. line 144-145 : In P. chilensis there is no variation. --> There was statistical significance.
9. table 3: -0.7 --> Correct to two decimal places
10. "However, at the species level, the average critical matric potential of P. chilensis was significantly higher than for the other species (1.200 vs. 1.010 154 kPa)." --> What is the 1.200 number?
11. A different post hoc test was performed for each table. I need an explanation as to why
12. "[30] evaluated 176 the germinative capacity in Bulnesia retama (a species that shares the northern distribution of Prosopis sp in Argentina) under different matric potentials, who together with [31] found that in matric potentials of -1.250 kPa, the germination capacity of B. retama is p"
--> Citation by author's name, not by number,,,Other places in the text need to be corrected.
13. line 190~ : ". On seed sources PO, QL and BL, there were a linear respons" --> BL --> LB
14. line 200~ : "Only in the seed source HJ was affected by the matric potential, indicating that the seed source of this specie could have im portance in the germination of this species with the use of conventional pregerminative treatments, [40] determi"
--> HJ isn't the only one affected by "matric potential". Other regions were also affected. Expression correction is required.
15. line 206 : "Even individuals possess adaptive traits [40]. " --> Is this sentence complete?
16. line 211 : "The germination capacity was high in seeds from the more arid source OV and LG." --> Is it true that germination capacity is high even in OV?
17. line 230 : IF --> FI
18. Re-edit the references as a whole. The scientific name needs to be edited in italics, journal name format, volume and issue number, etc.
Author Response
Response to the editor
We want to thank the reviewer for the detailed revision of the manuscript. We paid careful attention to the suggestions and made the corrections accordingly. We believe that the changes have greatly improved the manuscript. Below, we provide a detailed response to each specific comment.
Also, it was done in the attached draft with track changes in yellow, and I included another editor's suggestions that were similar to yours, so you can see them.
- introduction : The plant growth and development depend on climatic factors such as temperature, precipitation, relative humidity, solar radiation, and CO2. Thus, climate change may have serious consequences [1, 2]
--> It is true that plant growth is affected by climate change. However, if it is stated in the very next sentence that it has serious consequences, it seems to be too generalized. it need an explanation as to why this is such a serious outcome.
Response: The plant growth and development depend on climatic factors such as temperature, precipitation, relative humidity, solar radiation, and CO2. Thus, climate change may have serious consequences, due to the increase in temperatures and especially the decrease in rainfall in quantity and frequency [1, 2]
- line 37 : under this scenarios --> Exactly what scenario does this mean?
Response: Moreover, under this scenario of climate change, the projections indicate substantial losses of diversity of flora in this area [3, 42].
- line 77-78 : According to [48], P. chilensis is considered a phre ~~
--> need to change to like "according to authors [48]" ~~
Response: According to Miranda et al. [48], P
- line 108-109 : ", except in the LB origin where it had a linear behavior (water content decreases and germinative capacity decreases)."
--> PO also showed a similar pattern to LB.
Response: the highest germination capacity was not obtained, but at -6 and -33 kPa, except in the LB, PO and QL seed sorce where it had a linear behavior (water content decreases and germinative capacity decreases).
- table 1 : There are two duplicate parentheses. delete it
Response: The change was made
- Change commas to decimal points in tables. Other tables also need corrections.
Response: The change was made
- In Figure 2 and Figure 3, add the botanical name. In other words, it needs to be supplemented so that readers can understand it just by looking at the title of the figures.
Response: The change was made
- line 144-145 : In P. chilensis there is no variation. --> There was statistical significance.
Response: The change was made
- table 3: -0.7 --> Correct to two decimal places
Response: The change was made
- "However, at the species level, the average critical matric potential of P. chilensis was significantly higher than for the other species (1.200 vs. 1.010 154 kPa)." --> What is the 1.200 number?
Response: are the values of table 4, I corrected this sentence (line 172-174) “However, at the species level, the average critical matric potential of P. chilensis was significantly lower than for the other species (-1200 in P. chilensis vs. -1036 kPa on average in Q. saponaria and C. alba)”
- A different post hoc test was performed for each table. I need an explanation as to why
Response: Because when data normality was not met, a non-parametric Kruskal Wallis test was used, explained in point 4.3
- "[30] evaluated 176 the germinative capacity in Bulnesia retama (a species that shares the northern distribution of Prosopis sp in Argentina) under different matric potentials, who together with [31] found that in matric potentials of -1.250 kPa, the germination capacity of B. retama is p"
--> Citation by author's name, not by number,,,Other places in the text need to be corrected.
Response: cough were corrected and remained this way. “Rodríguez-Rivera et al. [30] evaluated the germinative capacity in Bulnesia retama (a species that shares the northern distribution of Prosopis sp in Argentina)”
- line 190~ : ". On seed sources PO, QL and BL, there were a linear respons" --> BL --> LB
Response: The change was made
- line 200~ : "Only in the seed source HJ was affected by the matric potential, indicating that the seed source of this specie could have im portance in the germination of this species with the use of conventional pregerminative treatments, [40] determi"
--> HJ isn't the only one affected by "matric potential". Other regions were also affected. Expression correction is required.
Response: The change was made in Line 117
- line 206 : "Even individuals possess adaptive traits [40]. " --> Is this sentence complete?
Response: The change was made, the sentence was deleted
- line 211 : "The germination capacity was high in seeds from the more arid source OV and LG." --> Is it true that germination capacity is high even in OV?
Response: If it is, has the highest values
- line 230 : IF --> FI
Response: The change was made
- Re-edit the references as a whole. The scientific name needs to be edited in italics, journal name format, volume and issue number, etc.
Response: The change was made

Round 2
Reviewer 1 Report
The authors have taken into account the suggestions of the reviewers and the article is acceptable for publication.
Reviewer 2 Report
review comments have been well applied.